# Soil and Phytomicrobiome for Plant Disease Suppression and Management under Climate Change: A Review

**DOI:** 10.3390/plants12142736

**Published:** 2023-07-23

**Authors:** Wen Chen, Dixi Modi, Adeline Picot

**Affiliations:** 1Ottawa Research and Development Centre, Science and Technology Branch, Agriculture and Agri-Food Canada, Ottawa, ON K1A 0C6, Canada; wen.chen@agr.gc.ca (W.C.); dixi.modi@agr.gc.ca (D.M.); 2Department of Biology, University of Ottawa, Ottawa, ON K1N 6N5, Canada; 3Univ Brest, INRAE, Laboratoire Universitaire de Biodiversité et Écologie Microbienne, F-29280 Plouzané, France

**Keywords:** agronomic practices, climate change, phytomicrobiome, plant pathogens

## Abstract

The phytomicrobiome plays a crucial role in soil and ecosystem health, encompassing both beneficial members providing critical ecosystem goods and services and pathogens threatening food safety and security. The potential benefits of harnessing the power of the phytomicrobiome for plant disease suppression and management are indisputable and of interest in agriculture but also in forestry and landscaping. Indeed, plant diseases can be mitigated by in situ manipulations of resident microorganisms through agronomic practices (such as minimum tillage, crop rotation, cover cropping, organic mulching, etc.) as well as by applying microbial inoculants. However, numerous challenges, such as the lack of standardized methods for microbiome analysis and the difficulty in translating research findings into practical applications are at stake. Moreover, climate change is affecting the distribution, abundance, and virulence of many plant pathogens, while also altering the phytomicrobiome functioning, further compounding disease management strategies. Here, we will first review literature demonstrating how agricultural practices have been found effective in promoting soil health and enhancing disease suppressiveness and mitigation through a shift of the phytomicrobiome. Challenges and barriers to the identification and use of the phytomicrobiome for plant disease management will then be discussed before focusing on the potential impacts of climate change on the phytomicrobiome functioning and disease outcome.

## 1. Introduction

As the global population is projected to reach 9.7 billion by 2050, and 10.4 billion by 2100 [1], food security can only continue to be a top priority for governments and international organizations, let alone climate change jeopardizing even more our ability to feed the world in sufficient quantity and quality. Pests and pathogens threatening plants greatly contribute to food insecurity since they are estimated to be responsible, each year, for the loss of 20 to 40% of the global crop production, representing an annual cost of USD 540 billion to the global economy [2,3]. At the same time, the increasing awareness of the harm caused by chemical pesticides to human, animal, and environmental health is paving the way to a thorough rethink of agricultural management practices in which the phytomicrobiome and its interaction with the environment has become a cornerstone. This new conceptual framework necessitates reconsideration of microbe–plant and inter-taxon interactions as a whole, shifting towards a more holistic perspective on the diverse roles and functions of microorganisms.

Microbiome research has rapidly grown as a widespread topic; the concept can be applied to diverse research areas, from medical science to food or marine science, forestry, aquaculture, and agriculture, and has even piqued the interest of industry and civil society [4]. In an attempt “for better coordination and collaboration across the fields of microbiome research”, Berg et al. [4] proposed a common definition of the term “microbiome”. The original definition, proposed by Whipps et al. [5], referred to “microbiome” as “the microorganisms involved but also encompasses their theatre of activity”, which has been considered the most comprehensive definition, according to an online survey and workshop discussions and was further detailed [4]. While “microbiota” is usually defined as the assemblage of living microorganisms in a specified habitat, the word “microbiome’ also includes a wide array of metabolites (such as signaling molecules, toxins, etc.) and microbial structural elements, including proteins and their building blocks—peptides, lipids, polysaccharides, nucleic acids, or mobile genetic elements [4]. In line with this definition, the phytomicrobiome can thus be defined as the microorganisms associated with any internal or external component of a plant, from the aboveground to the belowground, as well as the results of their activity. From an anthropogenic viewpoint, the phytomicrobiome includes beneficial members providing critical ecosystem goods and services and pathogens threatening food safety and security.

In terms of applications, two broad phytomicrobiome-based approaches that may offer alternative strategies for plant disease suppression and management can be defined (Figure 1), depending on the necessity to cultivate microorganisms before application in the field [6]. Indeed, microorganisms can be directly introduced as inoculants, either alone or as a mixture, to the ecosystems. Globally, the agricultural biocontrol market, including microorganism-based products, is valued at ~USD 6.6 billion with 39.5% attributable to North America. It was also predicted to reach USD 13.7 billion by 2027, based on a report published in 2022 (https://www.researchandmarkets.com/reports/5656078/, accessed on 8 March 2023). Alternatively, plant diseases may also be mitigated by in situ manipulations of resident microorganisms through agronomic practices without the need for industrial-scale production of microorganisms, such as those targeted in “conservation agriculture”. It relies on three principles: (i) minimal soil disturbance or absence of tillage, (ii) continuous soil cover with crops, cover crops, or a mulch of crop residues, and (iii) crop rotation [7]. While conservation agriculture primarily focuses on farming practices and not on breeding, it is worth noting that microbiome-assisted plant breeding, which involves selecting plants based on their enhanced capability to recruit beneficial microorganisms, can be considered within the second category from a broader perspective [8]. This has notably become possible thanks to the identification of specific genetic loci controlling such traits [8]. While soil amendments with compost, biochar, or even manure are primarily used for fertilization, the modified soil microbiota may indirectly impact plant protection against pests and diseases. In addition, at the frontier of these two approaches, the use of rhizo-microbiome transplants (RMT), although still in its infant stages, has recently shown promising results [9]. In this study, transplanting the rhizobiome of eggplant-resistant donors to tomato-susceptible recipients under greenhouse conditions resulted in a reduction of bacterial wilt disease incidence, with reductions of up to nearly 50%. It is crucial to consider the distinction between these two broad approaches based on the phytomicrobiome since their implications in terms of challenges and constraints may differ. Figure 1 provides a concise overview of these phytomicrobiome-related agricultural practices, which contribute to the suppression and management of plant disease either directly or indirectly by improving the overall agroecosystem health.

In this review, we will first gather relevant literature demonstrating how agricultural practices influence the phytomicrobiome and how these changes, in turn, affect ecosystem health. This includes improvements in soil health, the depletion and/or mitigation of diseases, and their overall impact. Additionally, we will address the challenges and barriers associated with identifying and utilizing the phytomicrobiome for plant disease management. Furthermore, we will explore the potential impacts of climate change on phytomicrobiome functioning, including plant pathogens, while emphasizing the implications for enhancing ecosystem resilience. While the advancements in this research field primarily target agriculture and cropping systems, their relevance extends to other domains such as forestry and landscape management in both urban and non-urban areas. Therefore, this review will encompass a broader scope and not be strictly limited to agricultural contexts, whenever relevant data are available.

## 2. Agronomic Practices, Phytomicrobiomes, and Plant Diseases

The phytomicrobiome plays a crucial role in supporting soil health, plant health and development, and mitigating plant diseases by signaling the root and modulating its rhizosphere microbiome accordingly [10]. The best example perhaps is suppressive soils that act as the frontline defense against specific or a broader spectrum of soilborne nematodes and pathogens [11]. The antagonistic ability of suppressive soils is owing to the diversity, makeup, and activities of indigenous microorganisms, which can be modulated by management practices such as tillage and crop rotation regimes, for example. Suppressiveness can be referred to as general or specific when a large or specific group of microorganisms is respectively responsible for the soil antagonistic ability [12]. For instance, the enrichment in *Pseudomonas* spp. was related to suppressive soils to apple replant disease, Fusarium wilt of tomato, Pythium damping-off of wheat, cucumber, Aphanomyces root rot of pea, clubroot diseases of Chinese cabbage, etc. [12]. Other functional guilds that may be responsible for soil specific suppression include *Flavobacterium*, *Chryseobacteria*, *Burkholderia*, and non-pathogenic strains of *Streptomyces* [13]. After World War II, many agricultural practices were implemented with the aim of maximizing yields and economic profit. However, these practices, such as the extensive application of pesticides and chemical fertilizers, have caused considerable harm to arable soils and their microbial inhabitants. Thus, sustainable agricultural practices are experimented with and introduced to restore soil fertility and suppressiveness to plant pathogens, by, at least in part, modulating the phytomicrobiome for the betterment of the overall agroecosystem.

### 2.1. Tillage

Tillage is a commonly used management practice for the containment of weeds, but it also plays a vital role in the even distribution of fertilizers and aeration in the soil. Conventional tillage (CT) is known to cause damage to soil aggregates and significantly alters the chemical properties of soil. This leads to reduced diversity and abundance of soil microbiota which in turn affect plant–root and plant–microbe interactions [14,15,16,17,18]. The Dust Bowl events in the US during the 1930s and in Canada between 1961 and 1990 serve as the most infamous examples of the consequences of extended tillage, which eventually caused wind-mediated soil erosion and resulted in nutrient-poor soils. To address these concerns, conservational agricultural practices, such as minimum tillage (MT) to no-tillage (NT), were implemented to minimize anthropogenic activities in farming and preserve microbial diversity and activity [19,20]. Compared to CT, both MT and NT have demonstrated positive outcomes, including enhanced colonization of arbuscular mycorrhiza that benefits plants through symbiotic associations, as well as increased soil carbon (C) and labile carbon, soil organic matter, soil moisture content, and microbial activity, fostering an intensified competition among soil microorganisms [21,22,23,24,25]. In addition, the adoption of long-term NT has been associated with increases in total nitrogen and available nitrogen compared to CT, as well as an augmentation of bacterial communities involved in nitrogen cycling [26,27]. For example, *Alphaproteobacteria*, *Betaproteobacteria*, and *Bacteroidetes*, comprised of plant growth promoting rhizobacteria (PGPR), were more abundant in NT as compared to CT [15,28,29]. Furthermore, compared to CT, soil bacterial communities under MT were significantly more diverse, and less conducive to the growth of *Fusarium graminearum*, a destructive pathogen of cereals, notably responsible for Fusarium crown rot or Fusarium head blight. These findings suggest that MT and/or NT can lead to increased soil fungistasis, although the direct impact on plant disease suppressiveness has not been thoroughly evaluated, limiting our ability to draw conclusive evidence on disease management [16,30]. Some studies have demonstrated that long-term NT or continuous application of MT can enhance soil suppressiveness in systems such as *Pythium ultimum*—*Lepidium sativum* (cress) and *Fusarium graminearum*—*Triticum aestivum* (wheat) [31,32]. However, other studies have found minimal influence of tillage practices on soil suppressiveness against *Rhizoctonia solani* and *Streptomyces scabies* in arable rotation systems in the Netherlands [33]. Moreover, NT coupled with other conservational agronomic practices, such as mulching, crop rotation, and the use of compost and organic fertilizers, often demonstrated effective suppression of diseases [34,35,36]. For instance, in minimum tilled fields cultivated in combination with crop rotation, suppression of take-all caused by *Gaeumannomyces graminis* var. tritici and Fusarium crown rot caused by *Fusarium pseudograminearum* and *F. culmorum* has been reported [36]. These examples highlight the potential of integrating multiple conservation practices to achieve enhanced disease management in agricultural systems.

### 2.2. Mulching

Mulching is used globally in agriculture to prevent moisture loss, maintain soil temperature, restrain weed growth, and prevent soil erosion. Yet, it exerts both positive and negative impacts on soil and plant health by influencing the composition and dynamics of associated microbiomes, including plant pathogens. Among different types of mulching, inorganic mulching, such as plastic film mulching (PFM), is particularly favored for its immediate economic benefits, such as increased crop yield, improved crop quality, reduced water usage, and lower inputs, especially in arid areas [37,38,39]. PFM, also known as soil solarization technique, promotes the formation of soil aggregates, leading to enhanced soil hydrothermal conditions, reduced evaporation, and improved nutrient availability. PFM also contributes to increased microbial activity in the soil and effectively mitigates soil-borne plant diseases [37,39,40,41]. Long-term mulching has been shown to enhance plant root growth, leading to an increase in colonization by Arbuscular Mycorrhizal Fungi (AMF) and a greater bacterial diversity. Concurrently, it contributes to a reduction in the population of phytopathogens, elucidating the role of PFM in disease suppressiveness [37,42]. Although PFM does not serve as a direct food source for microorganisms, it effectively hinders the spread of pathogens by physically blocking spores and other pathogen propagules from reaching the plants. Additionally, PFM enhances the crops’ resistance to diseases through healthy growth, facilitated by improved soil moisture and temperature [38,42,43,44,45]. Furthermore, silver plastic film or UV reflective plastic film, which reflects short wave UV light, offers further benefits in pest management by confusing and repelling insect pests like whitefly and aphids, which are known carriers of plant viruses. These types of plastic films, therefore, can effectively reduce the incidence of insect vector-mediated transmission of plant diseases, compared to black and yellow colored film which tend to attract pests such as green pea aphids and striped and spotted cucumber beetles [46,47,48].

However, it is important to acknowledge the negative impacts of plastic mulch. One of these is the accelerated biodegradation of Soil Organic Matter (SOM), leading to increased carbon/nitrogen (C/N) metabolism and eventual depletion of soil nutrients, particularly carbon [38,49]. Additionally, the residues of plastic film create a new microbial community niche known as the “plastisphere,” which can harbor potential pathogenic organisms like *Fusarium oxysporum*, *Alternaria alternata*, and *Didymella glomerata*. The use of plastic films also raises concerns about the issue of microplastic pollution in agroecosystems that needs to be addressed [38,50,51,52,53]. To address these concerns, biodegradable plastic films, such as starch-based polymers, have been introduced. These films can be easily degraded by soil microbes. While these environmentally friendly alternatives have shown similar success in crop productivity, comprehensive research is needed to understand their long-term implications for soil microbial communities and soil quality. It is crucial to assess potential risks for the soil, biota, and society [38,54,55].

An alternative to using inorganic plastic mulches is the utilization of organic mulches, such as straw, wood chips, or leaves. Organic mulching not only improves soil physical properties but also contributes essential elements such as carbon, nitrogen, and phosphorus to the soil, enhancing the nutrient uptake by plant roots [49,56,57,58,59]. At initial stages, plant residues in organic mulches readily decompose, enriching the soil with a nutrient pool and promoting the activity of bacterial communities involved in the decomposition, denitrification, and nitrification processes [60,61,62]. Organic mulches can maintain a favorable soil environment for plant development and growth, which is less conducive to pathogen proliferation and disease establishment. Particularly, mulches can help by modulating soil temperature and moisture levels, reducing the landing of aphids (virus carriers) and the fungal spores splashed by rainwater, and thereby potentially decreasing the incidence of certain diseases [43,46]. Percival [63] reported that mulches derived from single species significantly reduced the severity of Phytophthora root rot lesion in horse chestnut (*Aesculus hippocastanum* L.) while promoting tree growth and vitality. However, the effectiveness of organic mulches may vary depending on the specific type of mulch used (mulch type-dependent). For example, Chalker-Scott [33] highlighted that organic mulches may contain antimicrobial metabolites or harbor antagonistic microorganisms that can combat pathogens. It is important to note that organic mulches can also harbor pathogens that can infect plants. Therefore, using disease-free mulches and avoiding mulches made from diseased plants is crucial to preventing the introduction or spread of plant pathogens [43].

While mulching can offer several benefits in promoting plant and soil health, it is important to recognize that other cropping systems, like crop rotation and soil amendments, can also contribute to disease suppression and nutrient-rich soil.

### 2.3. Monoculture and Polyculture

The monocultural farming system, in the context of this review, refers to continuous monoculture that grows a single crop over multiple growing seasons on the same field. By contrast, polycultural systems, such as intercropping and crop rotation regimes, grow two or more types of crops at the same time or in successive years, respectively, on a given field. Intensive continuous monoculture can be a fundamental problem in field farming systems, horticulture, and agroforestry. As the global population increased, farmers tend to replace intercropping and crop rotation with monocultures of high-value cash crops, seeking higher yields, higher profits, and lower management costs [64,65]. However, long-term monocropping has had detrimental effects on soil health and increased plant diseases by disrupting the micro-ecological environment of the soil. The negative plant–soil feedback persists over years, leaving a legacy of soil with low Soil Organic Carbon (SOC) and nitrogen, while harboring an abundance of saprotrophic and pathogenic microbes, ultimately leading to reduced yields and nutrient-poor soils [66,67,68,69]. For instance, Zhao et al. [70] documented a decline in the relative abundance of potentially beneficial microbes like *Nitrospira* and *Trichoderma*, accompanied by decreased soil pH and organic matter content, as well as increased soil electrical conductivity (EC) and salt stress due to continuous cropping of coffee plants. Similarly, a study by Chen et al. [71] found that long-term monocropping of peanut led to the accumulation of pathogenic fungi such as *Acrophialophora levis*, *Aspergillus corrugatus*, *Aspergillus niger*, *Emericellopsis minima*, *Fusarium solani*, *F. oxysporum*, *Neocosmospora striata*, *Scedosporium aurantiacum*, and *Thielavia hyrcaniae*.

However, in some cases, a continuous monoculture of susceptible hosts can paradoxically trigger specific suppression against the causal agent following a major disease outbreak. An interesting example is observed in the context of continuous monoculture of barley and wheat, where a phenomenon termed “take-all decline” (TAD) has been observed following severe take-all outbreaks. In TAD, the pathogen, *Gaeumannomyces graminis* var. tritici, can be specifically suppressed due to the enrichment of diverse fluorescent *Pseudomonas* spp. capable of producing antimicrobial 2,4-diacetylphloroglucinol (DAPG). It is worth noting that the establishment of such suppressiveness in soils, such as those for controlling potato scabs, may require more than a decade of continuous monoculture. To sustain this suppressiveness, it is crucial for the responsible antagonistic microorganism strains to have a strong affinity for crop roots, facilitating their colonization, while the crops themselves can recruit and maintain such a symbiotic relationship.

Farmers have long relied on intercropping as a pest control strategy, but the mechanism by which it contributes to disease mitigation remains to be elucidated. A notable outcome of intercropping is the complex shift in the soil microbiome, leading to increased diversity, abundance, and activity in intercropped plots compared to monocultured plots. For example, legume cover crops coupled with zero tillage, or NT, favor AMF colonization [72], which can effectively counteract negative plant–soil feedback and offer numerous beneficial traits in the context of pathogen control. For instance, AMF enhances plant resistance by stimulating plant defense mechanisms during pathogen attack [40,73]. The push-pull method of intercropping has been successful in suppressing maize diseases, such as ear rot and maize kernel infections caused by *Fusarium verticillioides* and *Aspergillus flavus*, resulting in reduced mycotoxins (fumonisins and aflatoxins) levels [74,75]. Intercropping systems facilitate carbon and nitrogen sequestration, leading to the development of diverse fungal taxa with various ecological functions, including mycorrhiza, endophytes, saprophytes, decomposers, and bioprotective fungi [74,75,76]. For instance, intercropping maize with *Atractylodes lancea* (a traditional Chinese medicinal plant) acidified the rhizospheric soil and promoted the accumulation of beneficial PGPR such as *Streptomyces*, *Bradyrhizobium*, *Candidatus* Solibacter, *Gemmatirosa*, and *Pseudolabrys*, benefiting the growth of *A. lancea* [77]. In a fennel–pepper intercropping system, five terpene substances discovered in the soil and root exudates of the fennel rhizosphere (d-limonene, estragole, anethole, gamma-terpenes, and beta-myrcene) were found to inhibit *Phytophthora capsici* [78]. Similar observations were made in maize–soybean intercropping, where maize root exudates, such as cinnamic acid, vanillic acid, ferulic acid, and *p*-coumaric acid, inhibited *Phytophthora sojae*, the causative agent of Phytophthora blight in soybean [79]. The intercropping of black pepper and vanilla showed a lower relative abundance of *Fusarium oxysporum* in the rhizosphere soil of vanilla plants, while potentially beneficial fungal groups such as *Trichoderma* spp. and *Penicillium* spp. exhibited a higher relative abundance [80]. The benefits of intercropping are not limited to agricultural systems. Forest ecosystems, for example, develop disease-suppressive soils when trees are planted in multiple species as compared to monoculture [81,82].

Crop rotation, similar to intercropping, can benefit the soil by promoting diverse microbial communities, thereby altering the rhizosphere microbiome compared to monoculture practices. The combination of conservation tillage (minimum till/no-till) and crop rotation is often practiced to reduce disease severity and harness the advantages of increased microbial diversity [34,35,36]. Through crop rotations, farmers can disrupt the life cycles of soil-borne pathogens associated with specific crops or crop genotypes, thereby limiting disease pressure. This approach helps to break down the populations of pathogens that may have accumulated in the soil, minimizing their impact on subsequent crops [83]. A study on wheat leaf blotch disease caused by *Pyrenophora tritici-repentis* revealed a decrease in disease severity and an improvement in yield after six years of rotation, particularly in systems utilizing NT and diverse crop rotations [84]. Therefore, the diversity of crops in a rotation scheme plays a crucial role in mitigating plant pathogens through beneficial allelopathy. Additionally, specific crop species in rotation exert influence by shaping rhizosphere microbiomes that are antagonistic to particular pathogens [85,86]. The choice of crops in a rotation can impact its effectiveness in promoting soil health and crop productivity. Grain legumes, for example, can fix atmospheric nitrogen and increase soil fertility, providing benefits to subsequent crops like wheat and improving their protein content [87]. Other crops, such as grasses, contribute to building soil organic matter and enhancing soil structure, which leads to improved water infiltration and retention. In devising a crop rotation scheme, it is crucial to consider the ability of soil-borne pathogens to utilize alternative hosts [88] and persist in a dormant state within the soil, as well as the disease response of various crops [89]. Incorporating non-host plants into crop rotations holds utmost importance in reducing yield losses caused by soil-borne diseases, especially considering that certain pathogens can survive in the soil over the long term independently of their favored host [90]. For example, severe Fusarium root rot in peas cultivated in continuous rotation in the Canadian prairies was associated with reduced soil microbial diversity and lower concentrations of beneficial bacteria and AMF [91]. Conversely, enhanced crop diversification and continuous cropping featuring multiple crops were found to enhance the abundance of antagonistic soil microorganisms. This, in turn, mitigated the impact of TAD on wheat by reducing soil pathogen populations, as discussed above. In general, it is recommended to incorporate at least three distinct crops into a crop rotation scheme to optimize soil health and enhance crop productivity.

Therefore, to improve general soil health, it is crucial to implement crop rotation with carefully selected crops that promote the growth of beneficial endophytic and rhizosphere microbial communities. This approach holds the potential to foster sustainable soils with improved nutrient levels and a phytomicrobiome that suppresses pathogens. If monoculture is favored over polyculture to achieve higher yields and profits, fertilization becomes an additional approach to meet the crop’s nutrient requirements while also maintaining microbial diversity through soil nutrient enrichment. As mentioned in the introduction, various soil amendment practices are also implemented to further enhance plant protection against phytopathogens.

### 2.4. Soil Amendments

#### 2.4.1. Fertilization

Conventional tillage and intensive monocropping practices have been found to deplete soil nutrients, resulting in nutrient-poor soils with lower levels of soil organic carbon and microbial diversity. Consequently, fertilization becomes a necessary practice in agricultural fields to replenish nitrogen and phosphorus levels and improve crop yields. However, the use of chemical fertilizers in continuous fertilization has well-known drawbacks, like deterioration of soil health, threatening animal and human health, and contributing to air pollution through the emission of greenhouse gases (e.g., N_2_O), as well as leaching of phosphorus and nitrogen from soil to groundwater and other waterbodies leading to eutrophication and dead zones in aquatic systems [92,93]. In fact, agricultural soil management accounted for more than 74% of emissions in 2020, with fertilizers being the primary contributor [94]. It is estimated that over 50% of nitrogen and 90% of phosphorus can leach from soil to groundwater, further exacerbating eutrophication concerns [95,96,97,98,99].

Sun et al. observed that long-term fertilization had a minimal impact on the phyllosphere microbiome (microbes inhabiting the surface of leaves) and root endophytes (microbes living inside plant tissues), while the soil microbiome (including bacteria, fungi, and protists) was more sensitive to such treatments, with protists experiencing the strongest impact [58,59]. More than 30% of reduced diversity of the phagotrophic protists, which are microbial predators, was observed under long-term fertilization and raised concerns about the potential loss of functionally important microbial taxa due to abiotic changes caused by fertilization [100,101]. However, the resilience of the phyllosphere microbiome to fertilization remains unclear due to limited research in this area. In both mineral and organic fertilized fields, copiotrophic bacteria (bacteria that thrive in nutrient-rich environments) outcompeted oligotrophs (bacteria adapted to low-nutrient conditions) due to the increased nitrogen input, but organic fertilization resulted in higher richness and diversity of the overall bacterial community [102]. Additionally, intensive fertilization suppressed the plant’s preference for associating with PGPR communities that aid in phosphorus solubilization and the production of indolic compounds [92]. Plants deficient in essential nutrients like nitrogen (N), phosphorus (P), and potassium (K) may be more susceptible to pathogens because their metabolic functions are impaired. However, an excessive supply of these nutrients can also increase disease incidence. Therefore, the use of inorganic/mineral fertilizers should be carefully monitored to maintain a balance that supports plant health while minimizing the risk of disease development. For example, Huang et al. [103] elucidated the molecular mechanisms underlying Nitrogen-Induced Susceptibility (NIS) in rice plants, demonstrating that high nitrogen fertilization leads to increased susceptibility to *Magnaporthe oryzae* infection, mediated by an over-expression of pathogenicity-related genes in the fungus. Similarly, Bonanomi et al. [104] tracked the occurrence of Tomato spotted wilt virus (TSWV) infection in tomato seedlings and found that the use of mineral fertilizers and fumigation led to a higher rate of disease incidence (>80%) compared to the use of organic amendments such as alfalfa straw and biochar which had the lowest incidence of the disease (<40%). By contrast, low nitrate supplies were shown to increase the tolerance of *A. thaliana* towards *B. cinerea* [105]. Several other studies have reported similar findings, indicating an increased disease incidence with nitrogen fertilization. However, further research is needed to better understand the specific nutritional conditions that influence pathogen growth [106,107,108]. Actually, Dordas [109] describes in his review that plant pathogens respond differently to nitrogen fertilization depending on their trophic modes. For obligate parasites, such as *Puccinia graminis* and *Erysiphe graminis*, an elevated nitrogen supply leads to an escalation of the infection severity. Conversely, diseases caused by facultative parasites like *Alternaria*, *Fusarium*, and *Xanthomonas* species show a reduction in infection severity with high nitrogen supply, although this is contradictory to the earlier observations with *B. cinerea* infecting *Arabidopsis.* Similarly, Sun et al. [110] mention that nitrogen fertilization increases the infection rate of biotrophic pathogens causing powdery mildew, downy mildew, leaf rust, stem rot, and rice blast disease, while the opposite effect is observed in necrotrophic pathogens causing take-all, grey mold, and leaf spot disease. Sun et al. [110] further highlighted that factors such as plant genotypes, developmental stages, and growth conditions like planting years can introduce confounding effects, leading to conflicting results in the association between nitrogen nutrition and the efficacy of pathogen infection in plants.

Tao et al. proposed that bio-fertilizers or organic fertilizers exert their disease-suppressing effects through various modes of action: (1) direct antagonism by fertilizer, (2) inducing resistance or defense mechanisms in plants, and (3) indirect antagonism by stimulating microbes antagonistic to pathogens present in the soil [111]. Biofertilizers, such as vermicompost, are considered the most sustainable agriculture practices due to their ability to enhance soil quality by increasing beneficial microbes in addition to nutrient composition [112]. Biofertilizers made from mature compost enriched with biocontrol agents like *Bacillus* spp. and *Trichoderma* spp. have been shown to suppress Fusarium wilt disease [113]. The use of biofertilizers can improve soil health by directly suppressing pathogens or by modifying the indigenous microbial communities. Changes in disease suppression observed after biofertilizer application may be attributed to both the introduced microbes directly inhibiting the pathogen and alterations to the soil microbiota.

Organic fertilizers, such as compost, manure, or slaughterhouse waste, such as bone meal and poultry litter slurry, are derived from natural sources and offer a wide range of essential nutrients to plants. In addition to improving soil aggregation, water retention, soil organic carbon, and overall soil health [114,115], organic fertilizers also promote enzyme activity, soil microbial activity, microbial diversity, and richness [16,116,117,118]. The use of organic manure has been found to decrease soil-borne fungal pathogens like *Fusarium* and improve soil properties by modifying the composition and diversity of soil fungal communities, thereby reducing the incidence of soil-borne fungal diseases in the long term [119]. While composted manure or plant residues can suppress plant pathogens like *Pythium*, their disease-suppressive effects are attributed to microbial antagonism or plant host resistance, as decreased disease suppressiveness was observed following pasteurization or sterilization of compost [120]. The chemical heterogeneity of composting materials, like vineyard pruning waste, can affect the microbial communities and activities, leading to the development of suppressive composts that contain a higher relative abundance of Ascomycota and fungal genera like *Fusarium* and *Zopfiella*, which may play a role in controlling Phytophthora root rot in pepper plants [121]. Compost also has the ability to increase soil pH when amended with peat, thereby suppressing several pathogens like *Fusarium* that prefer acidic soils [122]. Green manure, especially from *Brassica* crops, can control weeds and alleviate plant diseases caused by pathogens in *Rhizoctonia*, *Verticillium*, *Sclerotinia*, *Phythophthora*, *Pythium*, *Aphanomyces*, and *Macrophomina* [123]. Many Brassica species produce glucosinolates, which hydrolyze into volatile toxins like isothiocyanates, acting as biofumigants and contributing to the disease suppressiveness of brassica-based green manure.

However, it is important to be aware that plant residue-based fertilizers often contain plant pathogens, while manure-derived fertilizers may harbor antibiotic resistance genes (ARGs) and mobilomes, which pose potential threats to human and animal health [124,125]. To mitigate the dissemination of ARGs, manure is often preprocessed before being applied to land. Recent reviews have shown that aerobic thermophilic and postdigestion composting could reduce >80% of ARGs, but different antibiotics may persist in the liquid (e.g., sulfonamides) or solid (e.g., tetracycline) phases of the manures and even after anaerobic digestion and composting [126]. Efforts should be made to address these concerns and develop strategies to ensure the safe and responsible use of organic fertilizers in agriculture.

#### 2.4.2. Treatment with Biofungicides vs. Chemical Pesticides

Pesticides, in their broadest sense, serve as a common tool used by farmers to manage and control plant pathogens and pests that can damage crops. When other agricultural practices, such as crop rotation and the use of resistant varieties, prove ineffective in controlling plant pathogens, farmers may resort to pesticides as a final measure to safeguard their crops and ensure optimal yields. Nevertheless, it is crucial for farmers to exercise caution and prudence in their pesticide use due to the potential negative consequences for both the environment and human health. The three concerns that any pesticide raises from overuse are the following: (1) Resistance: overuse of pesticides can lead to the development of resistance in targeted pathogens or pests, diminishing the effectiveness of the chemicals over time. For example, *B. cinerea*, the causal agent of grey mold, is considered a “high risk” organism able to rapidly develop resistance after the introduction of new fungicide classes, notably thanks to mechanisms based on drug efflux transport or by altering the target site(s) [127]. Resistance to quinone outside inhibitors (QoI, also generally referred to as strobilurins) and/or azole fungicides has also been reported to be caused by single and/or multiple mutations at the target site of *Plasmopara viticola* (the causal agent of grape downy mildew) or *Zymoseptoria tritici* (the causal agent of Septoria leaf blotch of wheat), as reviewed in Hawkins et al. [128]. This resistance can necessitate increased pesticide application, which in turn can have detrimental effects on soil health, human well-being, and the overall environment [93]. (2) Reduced fertility: pesticides can negatively impact soil fertility by harming beneficial microorganisms responsible for maintaining soil health and nutrient levels [129,130]. (3) Pesticides can also have adverse effects on the diversity of microorganisms in the soil, resulting in a loss of biodiversity and a decline in the overall health and productivity of the ecosystem.

To mitigate the potential disruption caused by pesticides, a combination of sustainable farming practices and the judicious use of pesticides is essential. By maintaining a diverse rhizosphere microbiome, these practices contribute to the resilience of the agroecosystem [131,132,133].

#### 2.4.3. Biochar

Biochar, a carbon-rich solid produced through pyrolysis, serves as a valuable tool for carbon sequestration and soil improvement [134]. One of the key mechanisms behind its beneficial effects is its ability to modify the microbial community in the soil. With its high porosity and large surface area, biochar provides an ideal substrate for microbial growth [135]. The soil amendment with biochar helps mitigate negative plant–soil feedback by altering the soil bacterial and fungal communities, while promoting the presence of beneficial bacteria like *Bacillus* and *Lysobacter* and suppressing plant pathogens such as *Rhizoctonia solani*, *Fusarium*, and *Ilyonectria* [136,137,138,139]. Biochar can also induce systemic resistance in plants by enriching the root microbiome with PGPR and fungi, reducing the plant’s vulnerability to soil pathogens as well as airborne pathogens [137,138,140,141,142,143]. While there are potential benefits to using biochar in agriculture, there are also concerns about its long-term safety and implications. The mechanisms by which biochar affects soil biota health are not yet fully understood or quantified. Additionally, the chemistry of biochar can vary depending on the feedstock and production conditions, making it challenging to predict the potential long-term consequences of its application [144]. Further research is necessary to gain a better understanding of the potential benefits and risks associated with biochar use in agriculture.

Biochar can also play a role in mitigating climate change by facilitating long-term sequestration of carbon, and influencing the fluxes of potent greenhouse gases in the soil, such as nitrous oxide and methane; therefore, it contributes to reducing emissions of these potent greenhouse gases [145,146]. Therefore, biochar has become a promising tool for sustainable agriculture, addressing both food security and environmental concerns.

#### 2.4.4. Chitin and Derivatives

Chitin, a polymer of N-acetylglucosamine, is the primary structural block in the exoskeletons of arthropods and fungal cell walls. Its derivatives, such as chitin, chitosan, and oligosaccharide derivatives, have found diverse applications, such as in treating wastewater from agriculture, industry, and urban household activities [147], removing phosphates and hydrosulfides from agricultural runoff, and being used as soil and fruit preservation additives to control crop pathogens [148]. These substances are known for their broad-spectrum antimicrobial properties against viruses, bacteria, fungi, oomycetes, and nematodes, and their ability to elicit defense systems in host plants [149]. Previous studies have demonstrated that chitosan can disrupt the development and growth of soil-borne pathogens, such as *Verticillium dahliae*, and post-harvest pathogenic fungi, including species of the genus *Colletotrichum* (which devastate thousands of plant species), *Botrytis cinerea* (causing grey rot on grapes), and *Rhizopus stolonifer* (causing fruit rot) [148,150]. These effects are achieved through mechanisms such as inducing cellular leakage, deforming cell walls, and impairing metabolic processes. Furthermore, chitosan has been found to stimulate the expression of bacterial chitinase and modulate the abundance of Actinobacteria and Oxalobacteraceae members, which play crucial roles in chitin degradation in the soil. This leads to improved soil suppressiveness against plant pathogens [151]. Interestingly, the application of crab shell (23% chitin) to infested soil demonstrated suppression of potato wart disease, a disease caused by an obligate soil-borne fungus, *Synchytrium endobioticum*, even though the population of resting spores was not directly affected [152]. This suppressive effect is likely attributed to changes in the soil microbiome, as temporary increases in the populations of soil nematodes and bacteria were observed. However, chitin-mediated management of potato wart disease was found to be less effective compared to crop rotations, which showed >75% reduction in resting spore populations when intercropping potato with rye (*Secale cereale*) and sunflower (*Helianthus annuus*).

#### 2.4.5. Clay Materials: Bentonite

Bentonite is a type of clay commonly used as a soil amendment in arid and semiarid regions. Bentonite is known to increase plant available water (PAW) by effectively retaining large amounts of water within its crystals, which contributes to improved plant growth and overall plant quality [153]. The water-holding capacity of bentonite has been shown to have a positive impact on soil microbial activity and nutrient cycling. By providing an optimal moisture environment, bentonite promotes the growth and activity of beneficial soil microorganisms, facilitating nutrient availability and cycling in the soil [153]. It serves as a valuable tool for soil detoxification due to its ability to absorb heavy metals like cadmium (Cd) and lead (Pb) from contaminated agricultural soils, thereby increasing the soil microbial population [154,155]. Soil amendment with bentonite can also influence fungal communities due to increased soil moisture retention and the formation of macro-aggregates [156]. These changes in soil properties can lead to reduced activity of phytopathogens such as *Alternaria*, *Bipolaris*, *Fusarium*, *Leptosphaeria*, and *Microdochium*. The increased competition from beneficial microorganisms may contribute to disease suppressiveness in the soil. Recent advancements in biocontrol techniques have led to the use of bentonite in the encapsulation of biocontrol agents like *Bacillus subtilis*, leading to improved survival chances of the biocontrol agents and enhanced activities against specific plant pathogens such as *Rhizoctonia solani*.

#### 2.4.6. Biocontrol Agents

Because the use of biocontrol agents to control pests and diseases has been extensively reviewed elsewhere [157,158,159,160,161], we will only provide a brief overview of this topic in the context of this review paper (Figure 2). First, it is noteworthy that biocontrol agents are defined differently by different authorities. In Canada, biological control agents are “insects, mites, nematodes, and other organisms used to control plant pests such as weeds or insects”, as defined by the Canadian Food Inspection Agency [157]. However, in France, biocontrol agents are classified into four types, including macroorganisms, microorganisms, chemical mediators (e.g., pheromones), and natural substances including plant defense elicitors [162]. Despite various definitions, biocontrol is a method to mitigate pest and plant diseases through effective use of living organisms or their derivatives, notably based on the use of microbial inoculants. Biocontrol agents help to alleviate plant disease through various mechanisms such as competition with pathogens; antagonistic activity by producing enzymes, toxins, and antibiotics; inducing plant systemic resistance; and/or direct lysis of the cell wall of the pathogenic organisms. Various microorganisms are used as biocontrol agents such as PGPR, endophytes, rhizosphere bacteria, and mycorrhizal fungi. Biocontrol agents can have host or pathogen specificity, whereas some of them, like PGPR, can be generally beneficial for all plants [163]. Indeed, as mentioned before, biostimulants (or biofertilizers), while primarily designed to enhance plant nutrition, are also known to have biocontrol properties, the two properties not being mutually exclusive. For instance, *Paenibacillus polymyxa* or *Pseudomonas fluorescens* are plant growth-promoting bacteria that have also shown interesting antagonistic properties against *F. graminearum* (Table 1) [164,165,166,167].

Researchers are recently investing in engineering phytomicrobiome to help the development of a healthy agroecosystem [168]. Perrone et al. suggested that resilience to climate change may become another criterion when selecting potential biocontrol agents [169].

Based on the comprehensive review of various agronomic practices discussed in this section, it is evident that several approaches, including NT or MT, crop rotation, cover cropping, and organic mulching, among others, have shown effectiveness in promoting overall soil health and enhancing disease suppressiveness (Table 2). However, it is crucial to acknowledge that the efficacy of these practices can vary depending on factors such as soil types, climate conditions, and crop genotypes. Additionally, while these practices can offer certain benefits in mitigating the impacts of climate change on plant diseases, they may not be fully adequate to address all the challenges arising from evolving environmental conditions. It is imperative to explore new and innovative approaches to plant disease management that can effectively tackle the challenges posed by climate change. One promising area of research lies in the utilization of phytomicrobiome-based approaches, including microbial inoculants and practices demonstrated in Figure 1, which leverage the intricate interactions between plants and their microbial communities to promote disease suppression and enhance crop resilience. However, as we will discuss in the next section, there are several challenges and pitfalls to the identification and use of phytomicrobiome-based approaches, which must be addressed in order to realize their full potential.

**Table 1 plants-12-02736-t001:** List of beneficial microbes with potential disease suppression activities.

Type of Organism	Organism Name	Used as	Targeted Disease or Pathogen	References
Bacteria	*Bacillus subtilis* MBI 600	PGPR, biocontrol agent	*Pythium aphanidermatum*, *Fusarium oxysporum* f. sp. *radicis-cucumerinum*	[170]
	*Bacillus velezensis* HN03	Biocontrol agent	Fusarium wilt (banana)	[171]
	*Bacillus thuringiensis* JCK-1233	Biocontrol agent, systemic resistance inducer	Wilt disease of pine caused by the nematode *Bursaphelenchus xylophilus*	[172]
	*Bacillus amyloliquefaciens* Group, *B. amyloliquefaciens*, *B. velezensis*, *B. nakamurai*, and *B. siamensis*	Biocontrol agents	Various important plant pathogens such as *Alternaria* spp., *Fusarium* spp., *Botryosphaeria* spp., *Botrytris* spp., etc. Extensive list is mentioned in the review cited here.	[173]
	*Paenibacillus polymyxa*	Biofertilizer, biocontrol agent	*Fusarium graminearum*	[164,165,166]
	*Pseudomonas* spp.	PGPR, biofertilizer, biocontrol agent	A broad array of phytopathogens	[167]
Fungi	*Ampelomyces* spp.	Biofungicide, biocontrol agent	Biocontrol of powdery mildews	[174]
	*Beauveria bassiana*	Biocontrol agent, entomopathogenic fungus	Various insect pests	[175,176,177]
	*Colletotrichum coccodes*	Mycoherbicide	*Abutilon theophrasti* (velvet leaf)	[178]
	*Coniothyrium minitans*	Biocontrol agent	*Sclerotinia sclerotiorum*	[179]
	*Metarhizium anisopliae*	Growth promoter, biopesticide	Biocontrol of insect pests	[177]
	*Trichoderma harzianum* ZC51	Biocontrol agent	*Fusarium oxysporum*	[180]
	*Trichoderma harzianum* SQR-T307	Biocontrol agent	Fusarium wilt of cucumbers	[181]
	*Trichoderma asperellum* T-34	Biocontrol agent	*F. oxysporum* f. sp. *lycopersici* race 1 causing Fusarium wilt of tomato	[182]

**Table 2 plants-12-02736-t002:** List of agronomic practices impacting phytomicrobiome and pathobiome.

Agronomic Practices		Impacts	
		Phytomicrobiome	Pathobiome
Tillage	Conventional tillage	Reduced diversity and abundance	No disease suppression observed
	No-Till and minimum tillage	Improved colonization of AMF and increased abundance of PGPR groups like *Alphaproteobacteria*, *Betaproteobacteria*, and *Bacteroidetes*, etc. in NT as compared to CT [15,28,29]	Increase of soil suppressiveness in systems such as *Pythium ultimum*—*Lepidium sativum* (cress) and *Fusarium graminearum*—*Triticum aestivum* (wheat) [31,32]
Mulching	Plastic film mulches	Long-term mulching increases plant growth and causes surge in AMF (Arbuscular Mycorrhizal Fungi) [37,42]	Physical barrier for pathogens, spores, propagules [38,42,43,44,45] Repellent for insect pests such as whitefly and aphids (carriers of plant viruses) and reduced incidence of insect-transmitted plant diseases [46,47,48]
	Organic mulches	Favorable to soil environments while providing nutrients to bacterial communities	Reduced severity of Phytophthora root rot with single species mulch [63]
Monoculture and polyculture	Monoculture	Decline in the relative abundance of potentially beneficial microbes (*Nitrospira* and *Trichoderma*), Decrease in soil pH and organic matter content Increase in soil electrical conductivity (EC), and salt stress [70]	Accumulation of fungi such as *Acrophialophora levis*, *Aspergillus corrugatus*, *Asergillus niger*, *Emericellopsis minima*, *Fusarium solani*, *Fusarium oxysporum*, *Neocosmospora striata*, *Scedosporium aurantiacum*, and *Thielavia hyrcaniae* in peanut long-term monocropping [71]. Occurrence of severe Fusarium root rot in peas cultivated in continuous rotation in the Canadian prairies associated with reduced soil microbial diversity and lower concentrations of beneficial bacteria and AMF
	Intercropping	Increase of carbon and nitrogen sequestration [74,75,76] Enrichment in diverse microbial taxa with various ecological functions such as mycorrhiza and/or endophytes, saprophytes, decomposers, bioprotective fungi or PGPR like *Streptomyces*, *Bradyrhizobium*, *Candidatus* Solibacter, *Gemmatirosa*, and *Pseudolabrys* [72,74,75,76,77]	Disease suppression: Decrease in maize kernels infections caused by *Fusarium verticillioides* and *Aspergillus flavus* along with reduced mycotoxins (fumonisins and aflatoxins) [74,75] Inhibition of *Phytophthora capsici* (likely attributed to the production of terpene in soil and root exudates) [78] Inhibition of *Phytophthora sojae* causing Phytophthora blight in soybean through maize root exudates such as cinnamic acid, vanillic acid, ferulic acid, and *p*-coumaric acid [79] Lower relative abundance of *Fusarium oxysporum* in the rhizosphere soil of the vanilla plants cultivated with black pepper along with higher relative abundance of potentially beneficial fungal groups such as *Trichoderma* [80]
	Crop rotation	Enrichment with diverse microbial communities	Decreased severity of wheat leaf blotch disease caused by *Pyrenophora tritici-repentis* with maximum impact in no-till and most diverse crop rotation systems [84]
**Soil Amendments**			
Fertilization	Chemical fertilizers	Resistance of the phyllosphere microbiome and root endophytes to long-term fertilization although soil microbiome (bacteria, fungi, and more importantly protists) was affected [58,59]. Out-competition of copiotrophic bacteria over oligotrophs	High nitrogen fertilization causing Nitrogen-Induced Susceptibility of biotrophic pathogens (e.g., powdery mildew, downy mildew, leaf rust, stem rot, and rice blast disease), and conversely, reduced infection by nectrotrophic pathogens (e.g., those responsible for take-all, and leaf spot disease) [103,110]
	Organic fertilizers	Increase in the richness and diversity of the overall bacterial community [113] Higher levels of soil microbial activities, microbial diversity, and richness [16,116,117,118] Shift in microbial community and activity in vineyard pruning waste, with higher relative abundance of Ascomycota and fungal genera such as *Fusarium* and *Zopfiella* (known to control Phytophthora root rot in pepper plants)	Decrease soil-borne fungal pathogens such as *Fusarium* [119] Suppression of *Pythium* by composted manure or plant residues [120] Alleviation of plant diseases caused by *Rhizoctonia*, *Verticillium*, *Sclerotinia*, *Phytophthora*, *Pythium*, *Aphanomyces*, and *Macrophomina* in addition to weed control with green manure, especially from Brassica crops [123] Suppression of Fusarium wilt disease with mature compost enriched with biocontrol agents like *Bacillus* and *Trichoderma* spp. [113]
Biochar		Decrease in the negative plant–soil feedback by altering the soil bacterial and fungal communities and augmenting the rhizosphere with beneficial bacteria such as *Bacillus* and *Lysobacter* [136,137,138,139]	Suppression of the plant pathogens such as *Fusarium*, *Ilyonectria*, and *Rhizoctonia solani* [136,137,138,139] Induction of plant systemic by enriching the root microbiome with PGPR and fungi [137,138,140,141,142,143]
Chitin			Suppression of both soil-borne pathogens, e.g., *Verticillium dahliae*, and post-harvest pathogenic fungi (e.g., *Colletotrichum* spp., *Botrytis cinerea* (grey rot on grapes), *Rhizopus stolonifer* (black bread mold causing fruit rot)) [148,150] Suppression of potato wart disease, caused by *Synchytrium endobioticum*, with the use of crab shell (23% chitin) although the population of resting spores was not directly affected [152]
Bentonite		Increase in soil microbial activity, nutrient cycling and/or shift in fungal communities thanks to the water-holding- and macroaggregate formation capacity of bentonite [153,156] Involved in soil detoxification by absorbing heavy metals like cadmium (Cd) and lead (Pb) from contaminated agricultural soils, thereby increasing soil microbial population [154,155]	Disease suppression of phytopathogens like *Alternaria* spp., *Bipolaris* spp., *Fusarium* spp., *Leptosphaeria* spp., and *Microdochium* spp., through increased competition of beneficial microbes [156] Use of bentonite in encapsulation of biocontrol agents like *Bacillus subtilis* for better survival chances leading to increased activity against *Rhizoctonia solani*

## 3. Challenges and Pitfalls to the Identification and Use of Phytomicrobiome-Based Approaches

Overall, the importance of the phytomicrobiome and their interactions for plant health is a widely acknowledged fact. Agronomic practices, as discussed above, are known to impact the functioning of the agroecosystem through a shift in the phytomicrobiome, and, as such, changes in agronomic practices can be a lever for plant disease mitigation. Although there are examples of success stories in which phytomicrobiome-based approaches have proven efficient in managing plant disease, many challenges, either conceptual, computational, or related to non-target effects, remain to be elucidated.

### 3.1. Conceptual Challenges

First and foremost, to effectively employ phytomicrobiome-based approaches, a profound and holistic understanding of the interactions between plants and their microbiomes is essential. Equally important is a comprehensive comprehension of the etiology, ecology, and life cycles of pathogens responsible for plant diseases that these approaches aim to address. Taken a step further, plant diseases should be viewed through the pathobiome concept rather than through the prism of the traditional one pathogen–one disease concept [183,184,185,186]. This latter concept, rooted in Koch’s postulate, implies that a plant disease is caused by a pathogen interacting with a plant host, ultimately leading to the expression of symptoms, whose magnitude depends upon the strain’s aggressiveness, the host’s susceptibility, as well as environmental conditions [186]. Yet, this concept may be too restricted to grasp the full picture of the disease process, as it fails to consider that host-associated microorganisms also work in synergy to cause disease, while their interactions with macroorganisms, primarily the host but also insects, may also foster disease outcome [184,187]. As defined by Bass et al. [184], the pathobiome is “the set of host-associated organisms associated with reduced (potentially reduced) health status, as a result of interactions between members of that set and the host”.

To illustrate the pathobiome concept in plant diseases, we can consider the Meloidogyne-based disease complex (MDCs). This complex is accountable for severe yield loss in major food crops worldwide and arises from the interaction between root-knot nematodes (RKN) and phytopathogenic fungi [188]. Initial infection by RKN leads to the apparition of root swellings and knots, which foster colonization by occurring fungal and bacterial communities, ultimately leading to necrosis and atrophy of the root system [188]. Other examples of such devastating collaborative work are tree diseases caused by complex interactions between the host, microbiota, and insects [189]. For instance, the Acute Oat Decline (AOD), which represents a major threat to native oak in the UK, is thought to be primarily caused by several bacterial species (notably *Brenneria goodwinii*, *Gibbsiella quercinecans*, and *Rahnella victoriana*) which have been consistently found concomitantly and abundantly in bleeding lesions or cankers while larval galleries of the bark-boring beetle, *Agrilus biguttatus*, are also often found in association with these lesions. Another example involves fungal wood pathogens of grapevine, responsible for Grapevine Truck Diseases (GTD), such as those causing Esca or Botryosphaeria dieback, which supposedly act synergistically with bacteria to foster disease development [190,191,192]. In particular, Bruez et al. [190] reported that the microbiota associated with symptomatic wood tissues of young grapevines exhibiting typical Esca symptoms (typical white rot necrosis) at an early stage were dominated by two plant pathogenic fungi (*Fomitiporia mediterranea*, known to be the primary causal agent of Esca, and *Phaeomoniella chlamydospora*) along with a few bacterial taxa (*Sphingomonas* spp. and *Mycobacterium* spp.). The relative abundances of the two fungal species range from 60% to 90% and 5% to 15%, respectively. The authors hypothesized that this fungal–bacterial association may work synergistically to increase wood degradation and the formation of white-rot necrotic tissues at the onset of Esca. The increased wood degradation ability was further confirmed using co-inoculation of *F. mediterranea* with a wood-isolated bacteria, namely *Paenibacillus* sp. under in vitro conditions, although it does not belong to the previously mentioned genera *Sphingomonas* or *Mycobacterium* [192].

Understanding the contribution of the phytomicrobiome as a whole to initiate, foster, or mitigate disease development remains complex but is crucial for plant disease management strategies. The development of phytomicrobiome-based strategies for plant disease control would thus require addressing the following questions: (i) Who are the members of the pathobiome and phytomicrobiome at taxonomic and functional levels? (ii) What is the nature of their interactions among themselves, with the plant host, and environment? and (iii) What is the ultimate impact on disease onset and progression?

To address these questions, numerous recent studies, as described in the first part, have focused on comparing the microbiota associated with symptomatic versus healthy tissues, with suppressive versus conducive soils, and/or investigating the microbiota shift under various agronomic treatments and/or upon pathogen infection before evaluating the overall impact on disease incidence and severity. These studies were primarily based on culture-independent methods and, more particularly, on metabarcoding, which allows for a more complete inventory rapidly (and at a relatively accessible cost) compared to culture-dependent methods and the now obsolete fingerprinting methods (e.g., Denaturing Gel Gradient Electrophoresis, Single Strand Conformation Polymorphism). To go further in exploring the interactions between pathogenic agents and the surrounding microbiota, co-occurrence network analysis based on metabarcoding data has gained growing interest over the past few years (e.g., [193,194,195]). These networks are based on correlations between pairs of taxa across multiple samples to identify pairs of taxa that are positively (i.e., co-present or co-absent) or negatively associated (i.e., mutually excluding each other) [196]. The common underlying hypothesis is that positive correlation between a pathogenic taxon and other taxa may help initiate and promote the disease, and as such, it can be a way to identify members of the pathobiome (that is, all taxa positively correlated to a pathogenic taxon, [194]). In contrast, negative correlations between a pair of taxa involving a pathogen may suggest competitive interactions and help identify putative biocontrol agents [193,194,195]. This may be too reductive since alternative biological explanations may account for positive or negative correlations [196]. For instance, it can be hypothesized that positive correlations may also be due to a prey–predator model involving, for instance, a fungal taxon and a mycophagous taxon, the more prey, the more predator, and vice versa. Alternatively, taxon aggregation may simply be due to dispersal limitations or similar niche requirements [195]. Nevertheless, reconstruction of microbial ecological networks provides statistical inferences for the interactions among keystone taxa within the microbiome-host holobiont through the disease progression process.

### 3.2. Computational Challenges

The development of metabarcoding, whether used alone or in conjunction with co-occurrence networks, has allowed researchers to gain a deeper insight into microbial interactions. However, it is crucial to recognize the limits and constraints associated with such an approach to prevent misinterpretation and to draw conclusions with appropriate caution. First of all, the current taxonomic resolution granted by the metabarcoding approach of 16S rRNA gene or Internal Transcribed Spacer (ITS) allows us to go down, at best, to the genus level, or, in limited cases, to the species level. To unequivocally identify pathogenic organisms, accurate taxonomic identification must be yielded at least to the species level, given that isolates belonging to the same genus (if not species) may be either phytopathogens, non-pathogenic, or could even be used as biocontrol agents [197]. For instance, *Pseudomonas* and *Erwinia* can be important phytopathogens, such as those responsible for the fire blight disease in Rosaceae plants (*Erwinia amylovora*) or brown rots (e.g., *Pseudomonas syringae*), while these genera are also known to include strains with biocontrol interest [167,198,199]. Soil-borne *Streptomyces* spp. also contain strains that are pathogenic causing potato scab, and non-pathogenic strains that suppress *S. scabies* through competing for nutrients or producing antibiotics [11]. For fungi, we can cite the example of strains of *Pythium oligandrum* which can be either pathogenic or antagonists to plant pathogens [200]. Likewise, atoxigenic strains of *Aspergillus flavus* (e.g., AF36^®^ or AflaGuard^®^, two commercially available products in the US market) are used as biocontrol agents to reduce aflatoxin contamination by *Aspergillus* spp. in maize, cotton, and/or nut crops [197]. In such cases, the distribution between beneficial taxa and pathogenic ones is quite impossible to achieve at the genus level, or even at the species level. Actually, development of additional barcodes down to the species level for some pathogens is possible, such as that designed by Cobo-Díaz et al. [201] targeting the translated elongation factor (TEF-1α) gene to identify *Fusarium* composition down to the species level [202]. In addition to TEF-1α, β-tubulin, RNA polymerase II second largest subunit (RPB2), and cytochrome c oxidase subunits (COI) are additional housekeeping genes that have been recognized as alternative secondary fungal DNA barcodes [203,204,205,206]. These genes have proven useful in determining species relationships within rust fungi [207]. Sequences targeting β-tubulin have also been reported to provide more accurate taxonomic assignment for *Penicillium* spp. (an important fruit spoilage mold) than ITS [208] although, to the best of our knowledge, a metabarcoding approach targeting these specific barcodes has yet to be developed. Taxonomic-level resolution granted by metabarcoding data can actually be assessed by generating a phylogenetic tree, notably based on Bayesian inferences, as performed in Belair et al. [209]. In their study, the authors constructed phylogenetic trees with Bayesian posterior probabilities (BPP) based on a local amplicon database completed with ASV sequences. This step allowed for the manual reassignment of ASVs to a more proper taxonomic rank, notably those returned by the pipeline as *Botryosphaeria dothidea* and *Neofusicoccum* spp. that were found clustered with a few other genera in the phylogenetic tree and therefore reassigned accordingly. A multi-affiliation output is actually provided by the FROGS pipeline to take into account such inaccuracies in taxonomic assignment [210].

Secondly, a common objective is to identify taxa that exhibit significant differences in abundance between a treated condition and a control condition. Yet, metabarcoding only allows data to be expressed as relative abundance, and, as such, is compositional [211,212]. As clearly stated by Gloor et al., “we cannot get information about the absolute abundances since this information is lost during the sequencing process”. An increase in absolute abundance cannot be inferred from one in relative abundance, contrary to what can sometimes be suggested. In other words, a taxon whose relative abundance is significantly increased upon pathogen infection does not necessarily mean that its growth is fostered by the pathogen and could actually be the exact opposite [211]. Rather, it simply means that there is an increase in proportion, as compared to the rest of the microbial populations. To gain access to absolute abundances, one possibility could be to estimate the whole population size and infer the population size of each taxon based on its proportion [212]. The size of bacterial populations could be estimated using classic microbiological approaches based on the enumeration of colony-forming units in Petri dishes. Yet, unlike metabarcoding, only viable and cultivable populations are considered with this approach. Also, results would originate from two different samples (i.e., two different biological samples would be needed, one for DNA extraction and metabarcoding and the other for microbiological enumeration). To overcome these issues, the use of digital droplet PCR, as an estimate of the absolute quantitation of the bacterial 16S rRNA gene and the fungal ITS marker copies in a DNA extract, could be an interesting alternative, especially because it does not need a calibration curve, as compared to classic qPCR [213]. However, it still represents an important bias due to the varying numbers of gene copies of 16SrRNA and ITS among taxa. Alternatively, Tkacz et al. [214] proposed a method involving the addition of a synthetic spike to the samples before DNA extraction. This spike contained small, known amounts of 16S rRNA sequences of *Escherichia coli*, serving as an internal standard. This approach allows for the correction of the initial microbial density, built upon which Zemb et al. further enhanced this methodology [215]. This method seems very convincing but is not yet widespread, probably due to the still early nature of publication. In addition, it would also need to be designed for microbial eukaryotes. Without any prior PCR amplification step, the recently developed long-read and single-molecule sequencing technologies, such as Nanopore or Pacbio SMRT sequencing, may also reduce such PCR bias while offering better precision in taxonomic assignment thanks to longer reads [216]. Regarding co-occurrence networks, only a few allow dealing with compositional data (e.g., SparCC and SpiecEasi) [217,218] and should therefore be considered. Beyond biological misinterpretation of correlations, as discussed above, one additional common error is to build co-occurrence networks based on a few samples, which is highly unreliable. Berry and Widder [219] recommended the use of 25 samples from similar environments to avoid species segregation simply because of habitat filtering. Best practices for co-occurrence networks are notably described in their paper, as well as in Goberna and Verdú [196].

Although metabarcoding data have allowed us to gain a more complete picture of the members of the phytomicrobiome and pathobiome, we are still limited in our understanding of how microorganisms interact to trigger or slow pathogenesis as well as foster or mitigate disease progression. The recent use of meta-omics technology, such as metagenomics, metranscriptomic, metabaolomic, or metaproteomic, in combination with culture-dependent analysis (or culturomics), may shed a new light on the underlying mechanisms of microbial interactions in the context of plant diseases. For instance, returning to the example of the AOD decline, Broberg et al. compared the metagenome, metatranscriptome, and metaproteome of inner bark tissues in AOD symptomatic versus asymptomatic trees in order to shed new light on the underlying functional mechanisms of lesion formation [220]. In addition to identifying functional genes involved in pathogenicity, they could also determine the preponderant role of *B. goodwinii* in lesion formation and discover two additional Gram-positive bacteria actively implicated in lesion formation. Gao et al. also carried out a quite comprehensive study by comparing the taxonomic and functional profiles of the microbial communities associated with Fusarium wilt-diseased and healthy chili pepper plants using metabarcoding and shotgun metagenomics approaches. Lastly, the utilization of synthetic communities, although a simplified system lacking the complete breadth of the phytomicrobiota, may still offer valuable opportunities to establish causal relationships and enhance our understanding of the individual or collective role of microorganisms and their impact on plant phenotypes [221]. This reductionist approach, involving the inoculation of a comprehensive microbial collection under gnotobiotic conditions, may aid in unraveling microbe–microbe and plant–microbe interactions [221].

### 3.3. Challenges Associated with the Application of Microorganisms in the Field and the Evaluation of Side Effects

After identifying inoculants with potential for mitigating plant diseases, a greater challenge arises in their transposition to production conditions and, most importantly, ensuring their successful colonization. To optimize the colonization and disease protection efficacy, considerations such as formulation, dosage adjustments, and strategic positioning of the treatments become crucial factors (see Qiu et al. [222] for a review of the new and emerging strategies to improve the survival and activity of microbial inoculants). These challenges primarily apply to approaches involving the introduction of microbial inoculants. In contrast, in situ manipulations of resident microorganisms through agronomic practices are, by nature, not affected by these concerns.

Another main issue is related to the ecosystem consequences of introducing microorganisms and their associated legacy effects. No one wants to risk playing the sorcerer’s apprentice by introducing microorganisms, which would lead to a disruption of the ecosystem’s functioning and ultimately cause more damage than the disease being treated. Recently, several reviews or opinion papers have tackled this issue [223,224,225,226]. Mawarda et al. [223] showed that, of 108 studies evaluating the impact of microbial inoculation on soil microbial communities, 86% showed that soil microbial communities were significantly modified after application of microbial inoculants, compared to control treatments, while the demonstration of their beneficial effects was actually not mentioned. Jack et al. [224] proposed a set of good practices based on Testing, Regulation, Engineering, and Eradication (TREE). The latter step included, as an example, the development of phages, as a ready-to-use antidote dedicated to eradicating microbial inoculants in case of unwanted invasion, although this seems rather inapplicable. Actually, the fact that the introduction of microbial inoculants led to a shift in microbial communities, which could persist up to 1 year after introduction compared to control treatments, is not problematic per se, as acknowledged by the authors themselves [223]. The extent to which these shifts, induced by the introduction of inoculants, were associated with disruptive functioning of the agroecosystem must be addressed if we want to fully evaluate the risks associated with this type of practice.

Also, the addition of manure or compost is known to induce shifts in microbial communities (see Section 2) while introducing a great deal of microorganisms to the ecosystem. Therefore, if this reasoning were to be followed through, should cautions associated with the introduction of “synthetic” inoculants also be applied to soil treatments with manure (either green or not) or compost? Yet, given their long history of use, to the best of our knowledge, no major direct microbial invasions have ever been associated with these practices (except perhaps for cyanobacteria blooms, which partly result from excess nutrients because of run off of fertilizer, notably from farmers). Likewise, there has been no reported evidence of microbial invasions after the introduction of microorganisms, although this may be difficult to detect because of the cryptic nature of microbes. Regarding unwanted invasions of introduced insects for pest control, ten cases have actually been reported over their history of use, according to Heimpel and Cock [227]. The authors argued that these cases have led to a focus on the risks rather than the benefits since the 1990s, leading to an improvement in the safety of biological control but also to a decline in the number of introductions. They proposed a framework and decision models that included both risks and benefits.

While we do not, by any means, suggest applying microbial inoculants without a scientific body of knowledge of their safety, we rather argue that microbial invasions and disruption of agroecosystem functioning caused by microbial inoculants, while being a possibility, are currently difficult to predict, and this should not be a reason to slow their development down. Beyond the risk of unwanted invasions, another important issue is the ability of microorganisms to produce, or to acquire the ability to produce, potentially harmful secondary metabolites which simply cannot be completely identified and characterized for technical reasons. As mentioned earlier, non-aflatoxin producing strains are being applied in the United States or in some African countries as a strategy to effectively displace aflatoxin producers in maize fields or nut tree orchards, leading to a reduced level of aflatoxin contamination [228]. However, concerns have been raised regarding the long-term effect of continuous application of biocontrol strains on the native population structure, including the risk of aflatoxin restoration in non-toxigenic strains through mating, considering the high heritability of aflatoxin production demonstrated in laboratory crosses and evidence for sexual recombination in the field [229,230]. Regardless, it is important to gather information on the mode of action, the potential of related species and strains to produce relevant metabolites/toxins, adverse effects observed in the (eco)toxicity tests, and the use of population genomics to help study the effect of biocontrol strains on native populations. These data are crucial for evaluating the associated risk [231]. In fact, in the latest consolidated version of the European Regulation 1107/2009 (as of 22 November 2022) concerning the placing of plant protection products on the market, a certain number of safeguards are demanded before approval of a product:(1)The microorganisms shall be deposited at an internationally recognized culture collection, and the species name of the microorganisms shall be identified unequivocally (no pathogens allowed).(2)The methods of analysis to identify and quantify them must be validated and shown to be sufficiently specific, correctly calibrated, accurate, and precise.(3)Their effectiveness in protecting plants from the targeted pests or pathogens must be demonstrated.(4)They shall not have any unacceptable effects on plants or plant products and on the environment, including fate and distribution in the environment, impact on non-target species, impact on biodiversity and the ecosystem. Risk assessment must fulfill data requirements for active substances, including microorganisms, as described in Regulation (EU) No 544/2011.

Overall, many challenges remain to be overcome. While we are just starting to grasp the phytomicrobiome functions and complex interactions within members and the plant host, another major issue for concern is climate change, e.g., global warming and increased incidence of extreme weather events, which is undoubtedly impacting the phytomicrobiome and the ecosystem services they provide.

## 4. Agroecosystem Resilience and Adaptation to Climate Change

### 4.1. Impact of Climate Change on Plant Pathogens

As highlighted in the introduction, pests and pathogens pose a significant threat to crop yields, contributing to food insecurity worldwide. This challenge is further exacerbated by the impacts of climate change, characterized by global warming and an escalating frequency of extreme weather events. These changes can directly impact crop physiology and productivity, while also indirectly influencing the phytomicrobiome, including plant pathogens [232]. Plant pathogens respond to changing climate in three key ways, as summarized in Table 3: (1) multiplication, whereby higher temperatures facilitate increased reproduction and subsequently lead to more severe disease outbreaks; (2) migration to new locations and host jumping; and (3) evolution, encompassing genetic trait alterations and speciation.

#### 4.1.1. Multiplication

Temperature and moisture are critical abiotic factors that affect the growth and reproducibility of pathogens. Even a slightly longer season can lead to the proliferation of disease propagules, triggering outbreaks. An example of this is the coffee rust disease epidemic that occurred in Colombia and Central America during the last decade. The epidemic was primarily caused by a reduction in diurnal temperature amplitude, which resulted in increased pathogen reproduction due to a shorter latency period [233]. The mean annual temperature has a significant impact on the abundance of soil-borne plant pathogens. Studies have shown that with increasing temperatures, the relative abundance of potential fungal plant pathogens in soils triples, turning the soil into a reservoir of infection [234]. Furthermore, climate change is predicted to indirectly increase the inoculum of *Fusarium* spp. in soils through increased cropping and warmer weather, leading to a higher incidence of disease [235]. Soil-borne pathogenic fungi such as *Fusarium*, *Pythium*, *Rhizoctonia*, and *Sclerotinia* are highly influenced by climatic factors throughout their lifecycle, particularly during the latency stage of infection, which is strongly dependent on temperature and moisture. Warmer winters, resulting in a shorter latency period, can lead to the emergence of more aggressive pathogens once favorable conditions for infection are present. For instance, *Fusarium* infects wheat during anthesis, especially when rainfall occurs, facilitating the dispersal of conidia from the soil to the wheat heads. Although hot and dry summers, particularly during anthesis, may not be favorable for Fusarium head blight, climate change has been projected to cause early anthesis, notably in the UK, due to warmer conditions and corresponding wet conditions. Thus, in turn, this leads to a severe epidemic of Fusarium head blight and an increased risk of mycotoxin production by *Fusarium* spp. [236].

On the other hand, it has been hypothesized that the unusually hot summers experienced in France since 2015 have contributed to the reduced severity of ash dieback caused by *Hymenoscyphus fraxineus* in French forests. These high summer temperatures were believed to be unfavorable for this pathogen [237]. However, the impact of climate change on plant pathogens is multifaceted, and not all pathogens will respond to temperature changes in the same manner. Therefore, it is essential to understand the specific responses of individual plant pathogens to environmental changes in order to develop appropriate management strategies for crop disease. Furthermore, as underlined by Madgwick et al., [238] it is necessary to incorporate both disease and crop models to more accurately assess the risk of disease epidemics in the future. This integrated approach allows for a comprehensive understanding of the interactions between pathogens and crops, enabling better predictions and more effective disease management strategies.

#### 4.1.2. Migration

The impact of climate change on pathogen populations can vary, with some experiencing range expansion or shifting while others may witness a reduction [239]. For instance, Battilani et al. [240] reported an increase in aflatoxin contamination in maize in Northern Italy and Eastern Europe over a fifteen-year period. Aflatoxin-positive samples have also been on the rise in France since 2015, attributed to unusually hot summers supposed to be more favorable to the growth of *Aspergillus* section Flavi [241]. While aflatoxin contamination is known to be frequent in hot regions such as Africa and Central-South America due to the thermotolerant nature of causal agents [242], climate change may amplify this contamination in previously aflatoxin-free countries such as Europe. Pathogens are also expanding their zones of infection and moving towards the poles as global temperatures continue to rise [243]. Similarly, the emergence of *Verticillium longisporum* in Canadian canola [244] was likely caused by unusually hot and dry summers in the Prairies that favored the proliferation and growth of the pathogen. In addition, increasing atmospheric CO_2_ levels can enhance plant productivity in the Nordic regions, but simultaneously increase the occurrence of crop pests and plant diseases [245,246].

Plant pathogens can be dispersed through various means, such as wind, water, rain, insects, or other vectors. However, the survival and spread of these pathogens are heavily influenced by environmental conditions and the availability of suitable hosts. One well-known example of a wind-dispersed pathogen is *Phakopsora pachyrhizi*, the fungus causing Asian soybean rust. This pathogen was spread by Hurricane Ivan in September 2004, moving from South to North America and affecting the world’s largest soybean-producing regions [247,248]. The long distance dispersal of fungal crop pathogens has been well documented by researchers [249]. To address the issue of fungal spore invasion into new territories with favorable disease conditions, the Borlaug Global Rust Initiative (BGRI) has established a Global Rust Reference Centre focusing on monitoring the presence and movement of rust fungi urediniospores, particularly for rusts infesting wheat [250,251]. In a pilot study conducted by Chen et al. [252], air spore samplers were used to observe the population of the air mycobiome at different locations in Canada along growing seasons. The study confirmed the presence of fungal plant pathogens, including rust fungi from the Ascomycota group, and smut fungi from the Basidiomycota group during rainy periods. Additionally, other studies have reported significant amounts of spores in dust, primarily composed of pathogenic fungi such as *Aspergillus*, *Cladosporium*, *Alternaria*, and *Penicillium* [253,254].

Similarly, soil-borne pathogens in genera *Fusarium* and *Sclerotium* face the risk of dispersal by rain splash during extreme weather events driven by climate change. Notably, *F. graminearum* and *F. culmorum* are known to produce macroconidia asexually, causing Fusarium head blight, which is predicted to increase by the 2050s due to early anthesis and conidia dispersal by rain [238]. Climate change also impacts arthropod vectors responsible for plant pathogen dispersal, leading to more aggressive diseases in plants due to changes in their reproduction and distribution patterns. For example, the outbreak of the mountain pine beetle, which is associated with blue stain fungi affecting pine trees in western Canada, has been linked to climate change [255]. Additionally, the western corn rootworm (*Diabrotica virgifera* virgifera LeConte), a carrier of maize chlorotic mottle virus, has exhibited an expansion pattern in Europe, possibly due to shorter and milder winters [256]. Another response of plant pathogens to environmental changes is host shifts or jumps, where new pathogens emerge following the introduction of host plants into new geographical locations. An infamous example is *Phytophthora infestans*, the causal agent of potato late blight, which triggered the devastating Irish famine as a result of host plants being introduced to non-native regions [257].

#### 4.1.3. Evolution

Abiotic factors play a significant role in the evolution and speciation of organisms, including plant pathogens. Changes in temperature patterns, such as warm winters and cool summers, can impact the lifecycle of a pathogen by reducing its latency periods. However, these shifts in environmental conditions can also lead to the emergence of new species or pathotypes that thrive in different thermal ranges compared to their parent species. One such example is *Botrytis sinoallii*, a new species first reported in China in 2010 that causes grey mold in *Allium* crops [258]. Another instance highlighting the adaptation of plant pathogens to warmer temperatures is the Ug99 race of *Puccinia graminis* f. sp. *tritici* (Pgt). Originating in East Africa, this race possesses virulence against Sr31, a common resistant gene in wheat. The Ug99 race is considered a significant global threat to food security due to its genetic variants with virulence against additional resistant genes such as Sr21, Sr24, Sr31, and Sr36, causing stem rust in previously resistant wheat varieties [259]. As a result, wheat stem rust epidemics, including in Europe, have been observed across five continents, with ongoing evolution driven by increasing global temperatures [251,260]. Similarly, the yellow rust fungus (*Puccinia striiformis*), has experienced the development of more aggressive and virulent strains adapted to warmer climates. The emergence of a diverse wheat yellow rust population has also been reported in the United Kingdom [253]. It has actually been revealed thanks to the use of an innovative transcriptomic approach based on the sequencing of infected leaves, which allowed us to circumvent the limitations of culturing such an obligate parasite [261].

Furthermore, it is also overly complex to forecast the impact of climate change on plant–pathogen interactions and disease outcome, especially because, as discussed before, it not only depends on the responses of the pathogens and the host, but also on those of the phytomicrobiome and how well these three components will adapt to changing climate conditions. In addition, attributing current plant disease emergence, expansion, or restriction to climate change is also challenging, as it depends on high-quality historical and current observational data, which is why disease surveillance is of paramount importance [239]. These findings suggest that climate change is likely to have significant impacts on pathogen populations and plant disease dynamics, requiring continued monitoring and adaptation in agricultural management strategies.

Among the newest innovation tools contributing to predicting epidemics, the genomic epidemiology approach, such as that developed by Chen et al. [252] and Hubbard et al. [261], has proven powerful to study migration pathways and may anticipate epidemic outbreaks in combination with global surveillance, accurate diagnosis, and the sharing of information through platforms (referred to as digital epidemiology or digital disease detection). In addition, plant disease sensing based on proximal and/or remote sensing, although still underdeveloped, may also help in detecting, monitoring, and forecasting plant diseases in the fields [262]. For instance, early detection of *Xylella fastidiosa*, one of the most dangerous plant bacteria widely spread in America but emerging in Europe, is a prerequisite for its eradication [263]. By comparing plant functional traits retrieved from airborne imaging spectroscopy and thermography with traditional molecular detection by qPCR, the authors identified spectral signatures of bacterial infection and were able to detect infection before the actual expression of visual symptoms.

**Table 3 plants-12-02736-t003:** List of plant pathogens influenced by climate change and their responses to climate change.

	Causes	Pathogen Names	Disease Name	Crops Affected	Countries Affected	More Comments	References
Multiplication	Due to increased temperature and humidity	*Hemileia vastatrix*	Coffee rust	Coffee	Colombia and Central America (2008–2013)	Increased temperature increased the pathogen population.	[233]
		*Fusarium graminearum* *Fusarium culmorum*	Fusarium head blight	Wheat	Global	Increased infection due to high abundance of conidia in soil and early anthesis of wheat.	[236,238]
Migration	Airborne	*Phakopsora pachyrhizi*	Asian soybean rust	Soybean	US	Hurricane Ivan caused the spread of spores leading to disease outbreak in the largest soybean-producing states.	[247]
	Insect-borne	Chlorotic mottle virus spread by western corn rootworm, *Diabrotica virgifera virgifera*	Necrosis	Maize	Europe	Western corn worm is a native American species and is invading Europe.	[256]
	New hosts	*Botrytis cinerea*	Blossom blight	Japanese plums	Chile	First report on Japanese plums in 2013. It infected plums in California in 1960.	[264]
	New location	*Aspergillus* section Flavi	Aflatoxin production	Maize	France, Europe	Originated in America and Africa and reported in France in 2013.	[240,241]
	New location Speciation	*Phytophthora infestans*	Late blight	Potato, tomato	Europe	Led to the Irish famine in 19th century.	[257]
		*Verticillium longisporum*	Verticillium stripe	Canola and other brassica crops	Canada, Europe	Moving polewards. Recently reported in Canada.	[244]
		*Botrytis sinoallii*	Grey mold	Allium crops	China	New species of *Botrytis* found in province of China in 2010 due to increasing temperature.	[258]
Evolution	New strains	*Puccinia striiformis* f. sp. *tritici* (Pst)	Stripe (yellow) rust	Wheat	Global	New strains Pst 1 and Pst2 are very aggressive and virulent. Strain adapted to higher temperatures with shorter latency period and increased spore germination percentage.	[251,260]
	New strains	*Puccinia graminis* f. sp*. tritici* (Pgt) Race Ug99	Stem (black) rust	Wheat	Global	Race Ug99 is the most aggressive that was reported first in Africa and is virulent to the resistant gene Sr31.	[259]

### 4.2. Phytomicrobiome Can Modulate Plant’s Response to Climate Change

Plants undergo many physiological changes in response to weather conditions, which can make them more susceptible to the pathogens that thrive under climate change, as discussed earlier. However, the phytomicrobiome, as previously mentioned, has the potential to contribute to plant adaptation and acclimation to climate change. Certain members of the phytomicrobiome, such as AMF and PGPR, can assist plants in mitigating the impacts of biotic and abiotic stressors [93]. The ability of these beneficial microbes to aid plants in coping with stressful events, including the warming and drought associated with climate change, can be attributed to several mechanisms. They can directly produce compounds that protect plants from desiccation, such as exopolysaccharides or 1-aminocyclopropane-1-carboxylate (ACC) deaminase, which counteract excessive levels of ethylene and enhance plant resilience [265,266]. Additionally, these microbes can enhance water and nutrient uptake, modulate root morphology, and act as regulators of stress-responsive genes. They induce the accumulation of plant osmolytes and antioxidants, promoting plant growth and stress tolerance [265,266].

Many plant growth-promoting microorganisms (PGPM), including strains like *Pseudomonas fluorescens* and *Bacillus subtilis*, have been successfully utilized for pathogen biocontrol, either as single strains or in consortia. These PGPM can effectively combat pathogens such as *Fusarium graminearum*, which causes wheat diseases [166]. Furthermore, PGPM are known to trigger induced systemic resistance (ISR) in plants, bolstering their defense mechanisms against attacking pathogens [267]. Mitigation of climate change can also be achieved by reducing N_2_O emissions from agricultural systems. AMF, renowned as plant growth-promoting fungi, can acquire ammonium and reduce N_2_O production. Additionally, the inoculation of N_2_O-consuming microbes can contribute to the alleviation of N_2_O emissions [268,269].

While PGPMs hold potential for mitigating plant stress induced by climate change, their effectiveness in the field can be challenging due to their unpredictable behavior. Soil warming, for example, can disrupt the functionality of PGPR, as a significant portion of their energy is diverted towards responding to elevated temperatures [270]. This diversion of energy can hinder their ability to provide beneficial effects to plants. Furthermore, warming causes changes in the flux of photosynthetic material belowground, leading to reduced colonization of AMF or a preference for AMF species requiring lower C [271,272,273]. This shift in belowground resource allocation can disrupt the symbiotic relationship between plants and AMF, potentially impacting plant nutrient uptake and overall resilience to climate-induced stresses.

Taken together, it is becoming increasingly evident that climate change is having a significant impact on plant pathogens and the phytomicrobiome. Elevated temperatures, changes in precipitation patterns, and increased carbon dioxide levels are leading to shifts in the distribution, abundance, and virulence of many plant pathogens, while also altering the composition and function of the phytomicrobiome. These changes can have wide-ranging implications for plant health, crop production, and ecosystem stability. Therefore, it is essential to further understand the mechanisms driving these shifts and develop effective strategies to mitigate their effects on plant health and agricultural productivity. This will require interdisciplinary collaborations between plant biologists, microbiologists, climatologists, and agronomists, along with innovative approaches to crop management and breeding.

## 5. Conclusions

This comprehensive and thorough review provides an exhaustive illustration of various aspects of phytomicrobiome research. A significant portion of our review is dedicated to introducing key concepts, covering different aspects of numerous challenges and pitfalls of phytomicrobiome research (such as the lack of standardized methods for microbiome analysis, the difficulty in translating research findings into practical applications, or the side effects of phytomicrobiome-based applications), as well as the impact of climate change on the phytomicrobiome’s functioning and disease outcome. We consider this review to be a valuable guide for agronomists, soil microbiologists, plant pathologists, and researchers seeking to develop a holistic approach for further research in this field.

We conclude that the phytomicrobiome is an essential component of soil and ecosystem health, and its potential to mitigate plant diseases and improve resilience to climate change is promising. However, identifying and utilizing the phytomicrobiome for disease management presents numerous challenges. Moreover, as climate change continues to affect global temperatures and precipitation patterns, the phytomicrobiome’s functioning and its interactions with plant pathogens may be altered, further complicating disease management strategies. Nevertheless, the potential benefits of harnessing the power of the phytomicrobiome are significant, not just for agriculture but also for other fields such as forestry and urban landscaping. Further research is needed to fully understand the mechanisms underlying the phytomicrobiome’s effects on plant disease and to develop practical management strategies that can be implemented on a larger scale.

## Figures and Tables

**Figure 1 plants-12-02736-f001:**
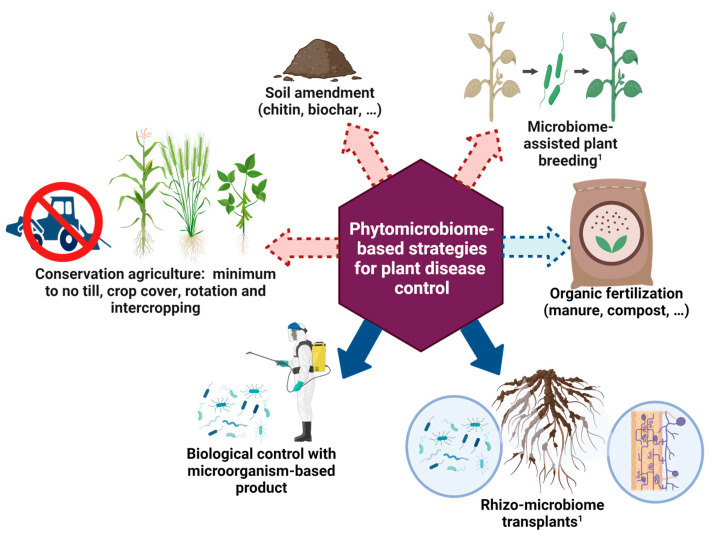
Outline of the phytomicrobiome-related agricultural practices that may lead to plant disease suppression and management. Red and blue arrows, respectively, point towards agricultural practices stimulating resident microorganisms or enriching soil with microorganisms either transiently or long-term. Plain and dashed arrows, respectively, point towards strategies that are directly or indirectly (notably through a betterment of the agroecosystem health) intended for plant disease control. Note: ^1^: novel emerging strategies but still at a low maturity level (in terms of Technology Readiness Level—TRL).

**Figure 2 plants-12-02736-f002:**
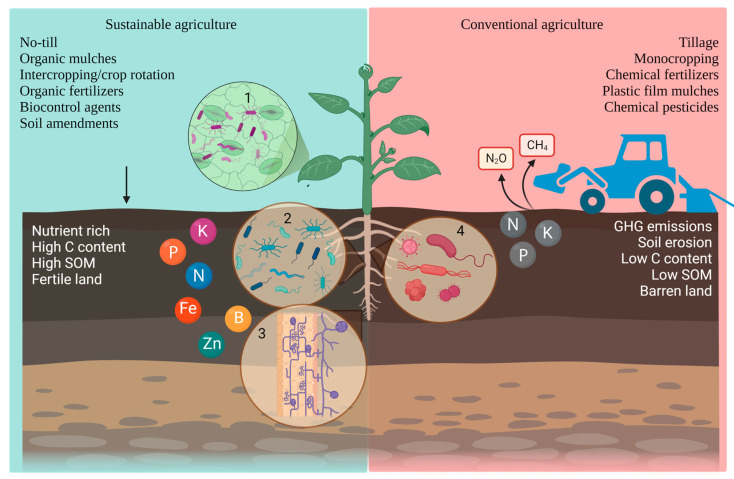
Illustration of the impact of agricultural practices on phytomicrobiome and plant pathogens. (1) Presence of endophytes in leaves (leaf microbiome); (2) Endophytes in root—root microbiome and rhizosphere microbiome; (3) Arbuscular fungi in the roots; and (4) Pathogens interacting with root. GHG: greenhouse gases; SOM: soil organic matter.

## Data Availability

No new data were created.

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
