# Peer review of "Soil and Phytomicrobiome for Plant Disease Suppression and Management under Climate Change: A Review"

_plants, 2023, doi:10.3390/plants12142736_

Round 1

Reviewer 1 Report

The review is interesting in terms of phytobiome and disease suppression management.

However, several things need to be improve before acceptance:

1. No detailed information about the microbial innoculant that suppress disease, their types and application.

2. Add table with detail information about types of beneficial microbes or beneficial phytobiomes that suppress disease with reference study.

3.  what is new in the interpretation of previously published papers i.e. what new in this paper brings to the field. Need to demonstrate in the manuscript.

Need to improve the language

Author Response

The review is interesting in terms of phytobiome and disease suppression management. However, several things need to be improve before acceptance:

RESPONSE: We thank the reviewer for his/her positive comments on our review article. We have carefully revised the manuscript according to his/her constructive comments.

  1. No detailed information about the microbial inoculant that suppress disease, their types and application.

RESPONSE: In Section 2.4.6, we have stated that the use of biocontrol agents to control pests and diseases have been extensively reviewed elsewhere, therefore, we decided to only provide a brief overview of this topic for the context of this review paper. Nevertheless, to address this comment, we have included a new Table 1, entitled “List of beneficial microbes with potential disease suppression activities” (new Ln630). In this table, we summarized a list of microorganisms that have been used as biocontrol agents or biostimulants, along with the plant diseases they have shown effectiveness in controlling.

  1. Add table with detail information about types of beneficial microbes or beneficial phytobiomes that suppress disease with reference study.

RESPONSE: We thank the reviewer for these relevant comments. A table summarizing microbial inoculant as well as their mode of application that are currently being used to control plant diseases is now provided (see Table 1)

  1. what is new in the interpretation of previously published papers i.e. what new in this paper brings to the field. Need to demonstrate in the manuscript.

RESPONSE: The originality of this paper and the contribution it brings to the field of phytomicrobiome fields is now underlined in the Conclusion section. We stated that “This comprehensive and thorough review provides an exhaustive illustration of various aspects of phytomicrobiome research. A significant portion of our review is dedicated to introducing key concepts, covering different aspects of numerous challenges and pitfalls of phytomicrobiome research (such as the lack of standardized methods for microbiome analysis, the difficulty in translating research findings into practical applications, or the side-effects of phytomicrobiome-based applications), as well as the impact of climate change on the phytomicrobiome’s functioning and disease outcome. We consider this review to be a valuable guide for agronomists, soil microbiologists, plant pathologists, and researchers seeking to develop a holistic approach for further research in this field” (Ln1197-1205).

Reviewer 2 Report

The content of the article is very rich and substantial. There are a few minor issues to note: 1) Two tables in the article. The first table 1 should be three lines. 2) The format of the references in the article should be consistent, and some documents need to be modified appropriately.

Minor editing of English language required

Author Response

The content of the article is very rich and substantial. There are a few minor issues to note: 1) Two tables in the article. The first table 1 should be three lines. 2) The format of the references in the article should be consistent, and some documents need to be modified appropriately.

RESPONSE: We thank the reviewer for his/her positive comments on the manuscript. The format of the references has been combed through to be consistent.  We would like to highlight that we have include three tables in this review, with a new Table 1, entitled “List of beneficial microbes with potential disease suppression activities”. The original Table 1 is now Table 2, which summarizes Section 2, entitled “Agronomic practices, phytomicrobiomes, and plant diseases”.  All three tables were formatted with three lines as suggested.

Comments on the Quality of English Language:Minor editing of English language required.

RESPONSE: The entire manuscript has undergone a thorough revision, resulting in improved logical flow and grammar corrections. Please see our submitted revision with track changes.

Reviewer 3 Report

In general, this is a well-written manuscript, as well as being an interesting review and in keeping with the theme presented, I also consider it the original work, however, it requires some minor changes before it can be published.

Changes to be made:

Lines 122, 152, 169, 321, 403, 739, 860, 861, 912, 1123, 1255, 1330, 1373, 1491, 1497, 1501,1510, 1513, 1578, 1579, 1584, 164 3, italicize Fusarium

Line 122, italicize Pythium

Lines 190, 423, italicize Phytophthora

In table 1 and 2, italicize Fusarium

In section 2.4.2. Treatment with Biofungicides vs. Chemical pesticides.

It is necessary to mention some examples of resistance to pesticides, and I also suggest mentioning some examples of microorganisms that have Phytostimulation and Phytobioregulation functions.

In section 2.4.2. 3.2. computational challenges

It is necessary to mention the most recent information on other more precise molecular markers for certain genera of phytopathogens, mention some examples of other molecular markers, and I also suggest mentioning the new phylogenetic analyzes through Bayesian trees, which are more precise for correct molecular identification. .

Lines 1435, 1544, italicize to in situ

In section 4.1.2. Migration

It is necessary to mention regarding the spectral indices, which can analyze physiological changes such as chlorosis, an important phenomenon for monitoring and diagnosis of pathogens of global importance. And also address the issue of remote sensing of pathogens by genomic analysis and dispersion of points in dynamic maps, to see how the migratory process of pathogens can be.

Conclusion

Adequate

Author Response

In general, this is a well-written manuscript, as well as being an interesting review and in keeping with the theme presented, I also consider it the original work, however, it requires some minor changes before it can be published.

RESPONSE: We thank the reviewer for his/her positive comments on the manuscript as well as his/her relevant comments.

Changes to be made:

Lines 122, 152, 169, 321, 403, 739, 860, 861, 912, 1123, 1255, 1330, 1373, 1491, 1497, 1501,1510, 1513, 1578, 1579, 1584, 164 3, italicize Fusarium

Line 122, italicize Pythium

Lines 190, 423, italicize Phytophthora

In table 1 and 2, italicize Fusarium

RESPONSE: Actually, we believe that when disease names are being used (e.g. Fusarium root rot), the genus name should not be italicized.

In section 2.4.2. Treatment with Biofungicides vs. Chemical pesticides. It is necessary to mention some examples of resistance to pesticides, and I also suggest mentioning some examples of microorganisms that have Phytostimulation and Phytobioregulation functions.

RESPONSE: We agree with the reviewer. Some examples of resistance to pesticides are now mentioning in this section while example of microorganisms with phytostimulation or phytobioregulation properties (or biofertilizers/biostimulants) are now included in the new table (see Table 1) as well as in section 2.4.6 Biocontrol agents. In particular, in Section 2.4.2, we added “For example, B. cinerea, the causal agent of grey mold, is considered as a “high risk” organism able to rapidly develop resistance after introduction of new fungicide classes, notably thanks to mechanisms based on drug efflux transport or by altering target site [128]. Resistance to quinone outside inhibitors (QoI, also generally referred to as strobilurins) and/or azole fungicides has also been reported to be caused by single and/or multiple mutations at the target site of Plasmopara viticola (the causal agent of grape downy mildew) or Zymoseptoria tritici (the causal agent of Septoria leaf blotch of wheat), as reviewed in Hawkins et al. [129]”.

In section 2.4.2. 3.2. computational challenges: It is necessary to mention the most recent information on other more precise molecular markers for certain genera of phytopathogens, mention some examples of other molecular markers, and I also suggest mentioning the new phylogenetic analyzes through Bayesian trees, which are more precise for correct molecular identification.

RESPONSE: We thank the reviewer for this comment. Recent information of more discriminant markers. We have added (Ln765-772) “Besides TEF-1a, β-tubulin, RNA polymerase II second largest subunit (RPB2), and cytochrome c oxidase subunits (COI), are additional housekeeping genes that have been recognized as alternative secondary fungal DNA barcodes [209-212]. These genes have proven useful in determining species relationships within rust fungi [213]. Sequences targeting β-tubulin have also been reported to provide more accurate taxonomic assignment for Penicillium spp. (an important fruit spoilage mold) than ITS [214] although, to the best of our knowledge, a metabarcoding approach targeting these specific barcodes has yet to be developed.”. We also summarized the use of Bayesian inference of phylogeny for more accurate taxonomic identification (Ln772-781).

Lines 1435, 1544, italicize to in situ

RESPONSE: Changes have been made accordingly.

In section 4.1.2. Migration: It is necessary to mention regarding the spectral indices, which can analyze physiological changes such as chlorosis, an important phenomenon for monitoring and diagnosis of pathogens of global importance. And also address the issue of remote sensing of pathogens by genomic analysis and dispersion of points in dynamic maps, to see how the migratory process of pathogens can be.

RESPONSE: We thank the reviewer for these comments. We have included a new paragraph related to remote sensing that may help monitoring and identifying plant pathogens as well as the use of genomic analyses to help identify migratory pathways of pathogens (Ln1130-1142).

Conclusion: Adequate

RESPONSE: Thank you.